# The Complex Interplay between Mitochondria, ROS and Entire Cellular Metabolism

**DOI:** 10.3390/antiox11101995

**Published:** 2022-10-08

**Authors:** Andrey V. Kuznetsov, Raimund Margreiter, Michael J. Ausserlechner, Judith Hagenbuchner

**Affiliations:** 1Department of Pediatrics I, Medical University of Innsbruck, A-6020 Innsbruck, Austria; 2Department of Visceral, Transplant and Thoracic Surgery, Medical University of Innsbruck, A-6020 Innsbruck, Austria; 3Department of Pediatrics II, Medical University of Innsbruck, A-6020 Innsbruck, Austria

**Keywords:** cellular metabolism, mitochondria, mitochondrial function/morphology, mitochondrial interactions, redox state, ROS, calcium, signaling

## Abstract

Besides their main function for energy production in form of ATP in processes of oxidative phosphorylation (OxPhos), mitochondria perform many other important cellular functions and participate in various physiological processes that are congregated. For example, mitochondria are considered to be one of the main sources of reactive oxygen species (ROS) and therefore they actively participate in the regulation of cellular redox and ROS signaling. These organelles also play a crucial role in Ca^2+^ signaling and homeostasis. The mitochondrial OxPhos and their cellular functions are strongly cell/tissue specific and can be heterogeneous even within the same cell, due to the existence of mitochondrial subpopulations with distinct functional and structural properties. However, the interplay between different functions of mitochondria is not fully understood. The mitochondrial functions may change as a response to the changes in the cellular metabolism (signaling in). On the other hand, several factors and feedback signals from mitochondria may influence the entire cell physiology (signaling out). Numerous interactions between mitochondria and the rest of cell, various cytoskeletal proteins, endoplasmic reticulum (ER) and other cellular elements have been demonstrated, and these interactions could actively participate in the regulation of mitochondrial and cellular metabolism. This review highlights the important role of the interplay between mitochondrial and entire cell physiology, including signaling from and to mitochondria.

## 1. Introduction

Mitochondria have been long time recognized as the powerhouse in various cells, particularly in high-energy consuming organs such as the heart, oxidative muscles, brain or liver. The energy production via oxidative phosphorylation (OxPhos) connects the oxidation of fatty acids or/and glucose with ATP synthesis from ADP, which is required for cell viability, survival and general cellular functions. In addition, mitochondria perform numerous other necessary functions and support many cellular pathways, participating, thus, in nearly all crucial metabolic processes. These organelles regulate cellular redox states, ROS and Ca^2+^ signaling (both acting as second messengers), produce important metabolites and are critically involved in apoptosis induction, autophagy and thermo-regulations [1,2,3,4,5,6,7]. Mitochondria are the sites of steroid hormone and porphyrin synthesis, the urea cycle, lipid and amino acids metabolism. They also play crucial roles in glucose sensing and insulin regulation [8]. The interplay between different mitochondrial functions in the cell, however, is not sufficiently investigated and can be strongly cell/tissue specific. Therefore, several lines of evidence demonstrated a tight link between mitochondrial functions and entire cellular metabolism. Moreover, mitochondria are able to monitor their surrounding environment, including intracellular ATP, as well as O_2_, ROS, Ca^2+^ and the presence of growth factors [9,10]. The existence of micro-domains with restricted diffusion, functional enzyme coupling and channeling could result in strong metabolic and mitochondrial heterogeneity [11,12]. Hence, each single mitochondrial environment can be significantly different from that of other mitochondria, potentially causing region-specific changes in the major mitochondrial properties and function.

In addition to their role in cellular bioenergetics, changes in the mitochondrial physiology, permeability, morphology and swelling are critical in cell fate decisions and injury [13,14]. The mitochondrial respiration coupled with the electron transfer through the electron transport chain are a major source of reactive oxygen species (ROS) production (see below). However, many open questions remain concerning the interplay between different ROS types and between ROS have originated from different sources. A central role of mitochondrial injury (with significant impairment of energy and cellular metabolism) has been well established in various diseases, such as inherited diseases, heart failure, ischemia-reperfusion injury, various myopathies, neurodegenerative diseases, diabetes, obesity and aging [14,15,16,17,18,19,20,21,22,23]. Interest in mitochondria was greatly renewed after the discovery of their important role in apoptosis induction due to the release of several pro-apoptotic factors [1,3,4,5]. Nevertheless, several questions and concerns remain and certainly need further investigation. This review summarizes and discusses previous studies on the crosstalk between the mitochondrial and entire cell physiology.

## 2. Mitochondrial ROS

Depending on the conditions, a few percentages points of the oxygen consumed by mitochondrial respiratory chain are reduced by electrons with the formation of superoxide radical (ROS) [24,25,26,27,28,29]. ROS then can be converted to hydrogen peroxide [30,31,32,33] by mitochondrial manganese superoxide dismutase (MnSOD), or by cytosolic/mitochondrial Cu,Zn-type superoxide dismutase (Cu, Zn-SOD) [34,35,36]. H_2_O_2_ in turn can be scavenged by peroxiredoxin 3 (Prx3) and glutaredoxin 2 (Grx2) [37,38,39,40,41], or by peroxisomal matrix enzyme catalase. Mitochondria represent, therefore, a major source of ROS in the cells. They themselves are also a very sensitive target for ROS with significant damaging effects. Mitochondria permanently produce ROS as a byproduct of respiratory chain transfer electrons to oxygen and its incomplete reduction. The respiratory chain complexes I and III (Figure 1) are considered the main producers of mitochondrial ROS in the form of superoxide [42,43,44], and then to H_2_O_2_, which in turn can easily escape mitochondria and be scavenged by the enzyme catalase in peroxisomes to water [45]. Hydrogen peroxide produced from superoxide radical is recognized now as one of the most effective cellular second messengers (such as Ca^2+^, see below) [46,47,48,49]. Mitochondrial ROS, particularly under pathological conditions, can damage several cellular elements such as DNA (and mitochondrial DNA), lipids of the biological membranes and various proteins/enzymes, especially by the oxidation of essential –SH groups [50]. In addition, a toxic modification of enzymes by ROS is the release of iron from Fe-S clusters. Oxidation of thiol groups thus may play a role in signaling [51] or be toxic, which depends on the oxidation reversibility. However, several lines of evidence also clearly demonstrate an important role of some (low) ROS concentrations in cellular signaling under normal conditions [46,47,48,49,52,53,54]. Therefore, mitochondria through ROS generation and redox-dependent signaling can control the general cellular metabolism and entire cell physiology, affecting cell differentiation, proliferation, survival and death (apoptosis) [10,24,54,55,56,57,58]. Importantly, dysregulation of this signaling system can be associated with various diseases. An important controller of mitochondrial superoxide generation rates is the mitochondrial inner membrane potential (Δψm). The uncouplers such as 2,4-dinitrophenol or FCCP and the uncoupling proteins (UCPs) may decrease mitochondrial ROS production that may have protective (e.g., cardioprotective) effects [29,59,60,61,62]. However, under some specific experimental conditions (ROS derived from alpha-glycerophosphate-dehydrogenase, or by complex I in reverse electron transferring), uncoupler FCCP may stimulate ROS production [63]. It is well known that uncouplers (CCCP or FCCP) significantly increase the respiration rate in a rather narrow range of concentration (1–5 µM) due to uncoupling, whereas at a higher concentration, they inhibit mitochondria. These effects can be cell/tissue specific. The mitophagy (removal of defected mitochondria) is mostly based on the low ΔΨm, and mitochondrial fission/fusion can be considered important parts of mitochondrial quality control [64,65,66,67]. The fusion allows mitochondrial matrix content exchange, whereas further fission produces normal mitochondria, providing, therefore, a repair process.

### ROS-Induced ROS Release and Mitochondrial ROS Flashes

Mitochondrial OxPhos and active respiratory chain usually operate concurrently with some ROS generation; however they can monitor their own elevated ROS—so, an interesting phenomenon “ROS-induced ROS release” (RIRR) has been demonstrated in several works [68,69,70]. Two modes of RIRR have been described, but in both, the ROS increase and dynamics can be biphasic—(1) gradual mitoROS increase, (2) followed by mitoROS flash [69,70]. MitoROS flash frequently occurs in parallel with mitochondrial depolarization and Ca^2+^ sparks in the same mitochondrion [70]. Since the RIRR phenomenon has been demonstrated under artificial conditions in isolated cells, physiological RIRR importance in intact organs remains unknown. However, it has been proposed that the complex ROS-ROS interactions and phenomenon of RIRR may be involved in the ROS signaling, as well as participating in the cellular network of redox signaling. Additionally, Wang et al., using the mitochondrial targeted superoxide probe, demonstrated the phenomenon of superoxide flashes in individual mitochondria in cells (cardiomyocytes) [71,72,73]. In these studies of mitochondrial ROS flashes in cardiac cells, however, caution must be taken because circularly permuted yellow fluorescent protein (cpYFP) not only presumably detects superoxide anion, but also changes in the cytosolic pH.

Since mitochondrial damage inhibits oxidative phosphorylation and increases ROS, heterogeneity of injury would be a result of spatial mitochondrial heterogeneity and thus heterogeneity of ATP and ROS production. Both local energy depletion and elevated ROS generation are damaging for mitochondria of the particular cellular region, causing, in turn, an increase in the extent of mitochondrial heterogeneity. ROS mito-flashes have shown to be always linked to mitochondrial depolarization (drop of Δψm), and mitoCa^2+^ sparks in various cultured carcinoma cells [70]. It has been demonstrated that ROS released from one single mitochondrion can initiate a ROS flash and depolarization of the same mitochondrion, or in the neighboring mitochondria. Using fluorescence imaging of mitochondrial flavoproteins, redox state wave propagation was first observed in cardiomyocytes under conditions of glucose deprivation [74]. However, despite numerous studies, the complex interrelationships between mitoROS flashes, mitoCa^2+^ sparks and ΔΨm are yet not fully understood. In particular, the interplay between ROS derived from different sources (mitochondrial respiratory chain, NADPH oxidases or xanthine oxidase), as well as the involvement of MPT, cellular and mitochondrial Ca^2+^ (mitoCa^2+^) in the mechanisms of ROS flashes initiations are still poorly identified. Additionally, the exact sequences of the events, such as ROS flashes, ΔΨm dissipation, and mitoCa^2+^ sparks, are still unclear due to technical problems and insufficient time resolutions, and have to be elucidated in the future. Moreover, the physiological significance of mitoROS flashes and these cellular waves is still not understood.

## 3. The Interplay between ROS and Ca^2+^ Signaling

Mitochondria actively participate in the cellular Ca^2+^ signaling (cellular second messenger) and homeostasis [75,76,77,78,79,80]. These organelles are now implicated in the control of many important aspects of cell physiology, such as calcium/ROS signaling under normal conditions [81], as well as in pathology, e.g., in ischemia-reperfusion injury [14]. Mitochondria also play central role in cellular Ca^2+^ [82] via the interplay between mitochondria, ER and cytosol. Under physiological conditions, Ca^2+^, as important signaling ion, can stimulate ATP synthesis, activating mitochondrial function via stimulation of various dehydrogenases, and thus activating mitochondrial and cellular energy metabolism [83,84,85,86]. Cytosolic Ca^2+^ levels then affect numerous other cell-signaling pathways. Ca^2+^ signaling is a main player used by mitochondria to modulate their activity for specific cellular demand. The structural and functional interactions between endoplasmic reticulum (ER) and mitochondria are extremely important in the regulation of cellular metabolism and functionality [87,88]. Ca^2+^ released from the ER can gain entry to the mitochondria, regulating enzymes’ activities and therefore the entire mitochondrial physiology. Changes in Ca^2+^ and mitochondria-ER signaling are associated with various stresses and pathologies (neurodegeneration, cardiovascular, immune diseases, etc.). Damage of the mitochondria and failure in OxPhos (ATP production) results in the disruption of Ca^2+^ signaling and Ca^2+^ homeostasis, leading to ER stress. The disruption of Ca^2+^ homeostasis often occurs concurrent with ROS production in response to various stresses, pointing to a strong link between redox signaling and intracellular Ca^2+^ handling/Ca^2+^ signaling [89,90]. Small mitochondria-ER contacts (at the micro-domain level) play important roles in cellular physiology, lipid/ion transferring, cell membranes dynamics and cellular signaling [91].

In addition, mitochondrial Ca^2+^ overload play a key role in several pathologies [92,93,94]. Under normal conditions, transitions between open and closed states of the mitochondrial permeability transition (MPT) pore operate to balanced/moderate release of mitochondrial Ca^2+^. It has been shown that mitochondrial Ca^2+^ overload is associated with the MPT pore opening, elevated ROS production, mitochondrial depolarization (ΔΨm loss), OxPhos uncoupling from respiration, mitochondrial swelling and cytochrome c release, as proapoptotic factor. The massive Ca^2+^ release from mitochondria results in cardiomyocytes hyper-contracture and cell death. MPTP opening results in cell death via necrosis, although whether the cell dies through apoptosis or necrosis depends on the ATP availability. Figure 2 shows fluorescence confocal imaging of mitoCa^2+^ elevation in HL-1 cells during photo-oxidative stress, observed by specific mitoCa^2+^ probe—Rhod-2. The application of various pro-oxidants (H_2_O_2_, organic peroxides, etc.) initiates similar effects. However, a complex interplay exists between ROS increase and increased mitoCa^2+^. While ROS increases the mitoCa^2+^ level, mitochondrial Ca^2+^overload may lead to elevated ROS generation and MPT pore opening [94]. Both effects can be prevented either by antioxidants, or by Ca^2+^ chelating [95]. It was also suggested that mitoROS, ATP and mitoCa^2+^ create a triangular system in which each may regulate the others [90]. However, due to the complex interrelationship between diminished cellular ATP and elevated mitoROS and Ca^2+^, the mechanisms and exact consequences of events are not completely clear. They can differ, depending on the physiological or pathological cellular status. Therefore, elucidating the molecular adjustments controlling this multifaceted triangle will continue to be a challenge in future. In some pathologies, the MPT pore opening is caused by high mitoCa^2+^ and other stimuli, including oxidants and/or depletion of adenine nucleotides [14]. These effects can be inhibited by acidic pH, antioxidants, e.g., reduced glutathione (GSH) and others or also by cyclosporin A. The link between Ca^2+^ entry and oxidative stress has been demonstrated as well in Parkinson’s disease, with a loss of small cluster of neurons (dopaminergic neurons) [96]. Additionally, mitoCa^2+^ overload, mitoCa^2+^exchange remodeling and metabolic dysfunction were associated with neuronal loss in Alzheimer’s disease progression [97].

## 4. ROS, Ca^2+^ and Various Kinases Signaling

The redox signaling plays a crucial role in the entire cell physiology. ROS can activate various protein kinases, such as PKA, ERK, PI3K, Akt, PKC, JNK, p38 [98]. It has also been demonstrated that the expression of wild-type or oncogenic RAF prevented excessive ROS production, mitoCa^2+^ elevation [95]. Protein kinase A is involved in the regulation of the NADH-ubiquinone oxidoreductase activity of complex I associated with reduced ROS production [99]. In contrast, increased mitochondrial ROS generation has been found for p66Shc protein [100]. H_2_O_2_ produced by p66Shc, localized in the mitochondria, oxidizes cytochrome c and activates MPT pore opening, promoting the induction of apoptosis. Taken together, these observations demonstrate that crucial mitochondrial pathways are a subject for the regulation via various cellular signaling mechanisms.

Mitochondria are considered a major source of ROS (see above). On the other hand, ROS production rates can be regulated by several upstream components (e.g., by small G protein Rac [101]). Intracellular signaling may regulate mitochondrial ROS generation as has been shown for several kinases, such as RAF, MKK6 or PKA [102]. A link between active RAF and mitoROS and mitoCa^2+^ changes have been recognized to be the events that precede the beginning of cell death by apoptotic mechanisms [95]. Therefore, RAF-MEK-ERK cell cascade activation/inhibition as well as AKT and Bcl-2 proteins may be involved in the elevated or inhibited mitoROS production. This has been shown also by applying siRNA or small molecular weight inhibitors, such as - LY294002, UO126 and/or BAY43-9006 [95].

In addition, an important role of the protein kinase C (PKC) [103] has been demonstrated through pro-apoptotic protein p66Shc, which translates oxidative damage into cell death by acting as a ROS, producer, in form of H_2_O_2_ [100]. The participation of several PKC isoforms during redox stress, with differences in their major biochemical properties, shows very complex patterns of general PKC signaling [103]. The activation of NADPH-oxidase and protein kinase C (PKC) under hyperglycemia and diabetic complications is associated with ROS production [104]. The oxidative stress, due to imbalance between the antioxidant system and ROS production in hyperglycemia (diabetes and its complications) is associated with the activation of PKC isoforms and the accumulation of advanced glycation end products [104,105]. Mitogen-activated protein kinases, including ETK, p38, and JNK (stress-activated protein kinase), are present downstream of the Src–PKC signaling system. This involves redox-sensitive transcription factors activation via PKC and tyrosine kinase. The reduced glutathione level steatotic liver is associated with increased ROS (oxidative stress), ER-cycling damage and the activation of JNK [105]. In addition, it has been shown that the hypoxia induces atrial fibrillation through the JNK/ROS pathway [106]. ROS or reactive nitrogen species activate JNK, which plays a role in apoptotic and/or necrotic cell death [107] (also after tumor necrosis factor treatment).

Mitochondria are well fitted to meet both the signaling and metabolic/bioenergetic cell requirements. Mitochondrial biogenesis and dynamics can be strongly linked to the ability of mitochondria to sense cellular energy status. One of the main enzymes for low-energy sensing in the cell is AMP activated protein kinase (AMPK) [108,109]. In muscles challenged with increased workload (e.g., training, endurance exercise), or various pathological circumstances (mitochondrial diseases, genetic defects, or defects in mitochondrial respiratory complexes), the proliferation of mitochondria acts as an adaptive response to decreased cellular energy levels. In other cells (e.g., neurons), energy sensing mechanisms may also control the directorial transport of mitochondria to the cellular regions of greater energy demands. Growing evidence demonstrates that AMPK can be also a critical controller of mitochondrial biogenesis [109,110].

## 5. Imaging Analysis of the Changes in Mitochondrial Redox State

Direct imaging of mitochondrial functional state in situ (in permeabilized muscle or cardiac fibers) and in intact cells is able to analyze mitochondrial NAD(P)H (two photon) and mitochondrial flavoprotein (confocal) auto-fluorescence. Both of these fluorescence emissions can be used for the sensitive assessment of the region-specific mitochondrial redox state [11,111]. The fluorescent flavoproteins and NAD(P)H intensities demonstrate inverse fluorescence signal behavior. Mitochondrial flavoproteins (the mitochondrial membrane integral components) are fluorescent only in an oxidized state and NAD(P)H oppositely only in a reduced state. This gives the possibility to continuously monitor changes in mitochondrial redox state upon additions of substrates, ADP, uncouplers or inhibitors. Representative Figure 3 demonstrates flavoproteins (Flavo) and NAD(P)H fluorescence (two photon excitation) in permeabilized cardiac muscle fibers (mitochondria in situ) and intact hepatocytes. In permeabilized cardiac muscle fibers (Figure 3), a transition from oxidized to strong reduced state of mitochondria can be seen after the addition of mitochondrial substrate, glutamate (Glu). In intact liver cells (Figure 3), a transition from some intermedium redox state to a significantly more oxidized state is visible after the activation of respiration by the addition of mitochondrial uncoupler 2,4-dinitrophenol (DNP). The additions of other mitochondrial substrates (e.g., octanoyl-L-carnitine), ADP and KCN complex IV inhibitor resulted in strong changes in the mitochondrial redox system visible from the changes in these two fluorescence emissions, and it can be quantitatively estimated (not shown). The quantification does not depend on the material amount since all changes can be calculated in percentages between fully reduced (KCN) and fully oxidized states (no substrates, saturated air oxygen).

This imaging approach established also the mitochondrial heterogeneity phenomenon [11], with observation of the mitochondrial subpopulations’ different properties. For example, a much higher oxidative state of subsarcolemmal as compared with intermyofibrillar mitochondria was observed [108]. The subsarcolemmal mitochondrial flavoprotein autofluorescence signal in skeletal muscle fibers (M. rat quadriceps) was found to be four times higher than intermyofibrillar auotofluorescence. The identification of membrane potential and mitROS revealed also pathologically altered mitochondrial heterogeneity (e.g., after cold ischemia-reperfusion and transplantation of rat hearts) [109].

The mitochondrial imaging, therefore, permits the assessment of mitochondrial defects topology, providing information about molecular mechanisms of various pathologies. Similarly, flavoprotein redox states have been demonstrated in intact cardiomyocytes under conditions of substrate (glucose) deprivation [76]. This included metabolic transients, well-coordinated redox transitions, and wave-like redox propagation within one cell and even between cells. The mechanism may involve some diffusible cytosolic signaling molecules, endogenous substrates, and oxygen. An imaging approach therefore permits the analysis of dynamic mitochondrial and cellular redox-state behavior. Importantly, flavoproteins and NADH fluorescence were fully co-localized with MitoTracker Green, a well-established fluorescent marker for mitochondria [110,111]. Although Mitotracker and flavoproteins have the same excitation/emission spectra, very big difference in the fluorescence signal (very low flavoproteins signal) allowed discriminating them. Moreover, flavoproteins fluorescence is auto-fluorescence of the integral mitochondrial components, whereas MitoTracker Green can be added later. So, the ratio of the intensities of fluorescent flavoproteins and NAD(P)H can be particularly useful, because it is non-sensitive to any other fluorescence types, thereby eliminating the possible side effects of artificial fluorescent probes.

## 6. Adaptive Changes of Mitochondrial Function/Morphology as Responses to the Changes in Surrounding Cytosol Composition

Numerous studies have shown that mitochondria and mitochondrial function may change as an adaptation response to condition/environmental changes.

Both mitochondria and the energy transfer networks may deteriorate under pathological conditions, such as decreased cellular ATP and increased inorganic phosphate levels. In contrast, in some cells, such as human carcinoma cells, mitochondrial function can be significantly improved and respiration (OxPhos) may be increased after a large decrease in the intracellular ATP level. This was demonstrated for cells treated with 2-deoxy-D-glucose (2-DG). This treatment inhibits the key enzyme, hexokinase, which is the first step of the glycolysis. This treatment leads to cellular energy depletion (~50% decrease in cellular ATP level was observed). Very recently, it has been shown that 2-DG treatment and ATP depletion resulted in significantly enhanced mitochondrial respiration and inner membrane potential, also with changes in mitochondrial morphology in the direction of more network organization [112]. The protein expression analysis demonstrated that 2-DG treatment activated AMPK (elevated pAMPK/AMPK ratio), increased mitochondrial fusion proteins (mitofusins 1 and 2) and decreased mitochondrial fission proteins (Drp1). This study therefore suggests a strong link between respiratory function and structural mitochondrial organization as a response to the energy status of the cell. The authors therefore proposed that the mitochondrial network functionality can be higher than the disconnected mitochondrial functionality [112].

## 7. Conclusions

In addition to the production of energy in the form of ATP, mitochondria play a crucial role in the entire cell physiology through their participation in many metabolic and signaling pathways. Mitochondria are able to monitor their surrounding environment, including intracellular ATP, oxygen, ROS, Ca^2+^, growth factors, etc. These organelles may change their properties in response to the changes in the cellular metabolism (signaling in). On the other hand, several factors and feedback signals from mitochondria may influence the entire cell physiology (signaling out). An interaction of mitochondria with cytoplasmic elements plays a causative role in the regulation of cellular metabolism and cell death. The many lines of evidence demonstrate the existence of a strong and undoubted interplay between the mitochondrial and entire cell physiology.

## Figures and Tables

**Figure 1 antioxidants-11-01995-f001:**
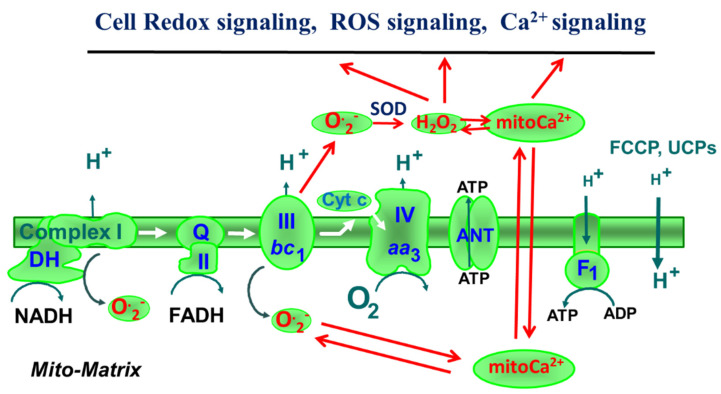
Interrelation between mitochondrial ROS, Ca^2+^ and cellular signaling. O._2_—superoxide radical; SOD—superoxide dismutase; UCPs—uncoupler proteins; mitoCa^2+^—mitochondrial Ca^2+^; Cyt c—cytochrome c.

**Figure 2 antioxidants-11-01995-f002:**
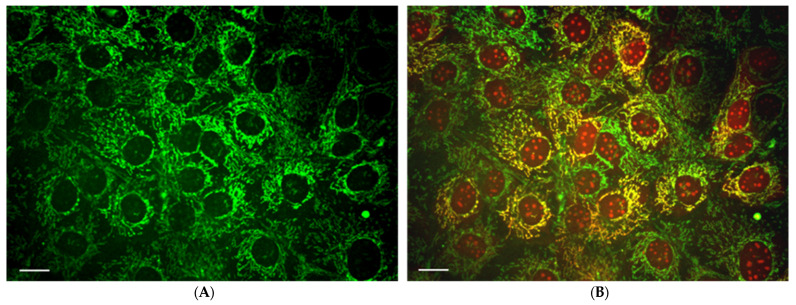
Representative images showing the increase in mitochondrial Ca^2+^ in photo-oxidative stress. (**A**) Mitochondria in HL-1 cells visualized with MitoTracker Green with low mitoCa^2+^. (**B**) Increased mitoCa^2+^, in these cells, during photo-oxidation (in time) can be detected by mitoCa^2+^—specific fluorescent probe Rhod-2 (yellow color in merge image (**B**). Scale bar 20 µm.

**Figure 3 antioxidants-11-01995-f003:**
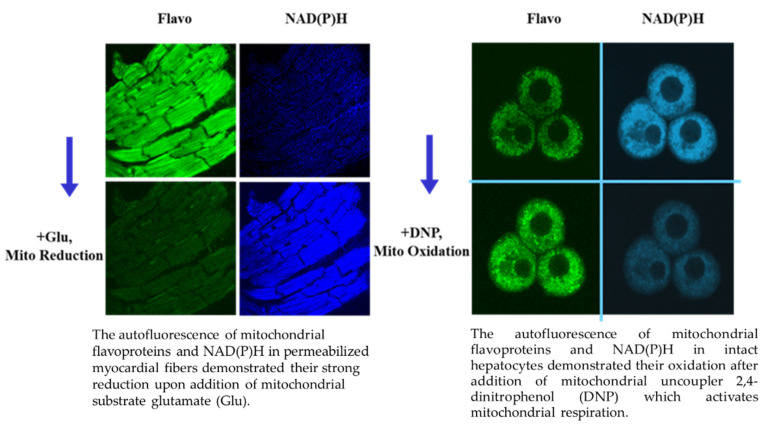
Changes in mitochondrial redox state visualized by fluorescent confocal (Flavo) and two photon (NAD(P)H) microscopy in situ and in vivo.

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
