# Peer review of "The Complex Interplay between Mitochondria, ROS and Entire Cellular Metabolism"

_antioxidants, 2022, doi:10.3390/antiox11101995_

Round 1

Reviewer 1 Report (New Reviewer)

The article considers the problems interplay between mitochondria, ROS, and entire cellular metabolism. But throughout the manuscript there are grammatical and syntactical errors. It is necessary throughout the text to correct and simplify sentence structures in which authors often use verbose introductory parts that are best placed at the sentence end. Authors often use introductory words (“However,”, “For example,” etc). This, in my opinion, complicates the text perception. Authors must carefully correct the text and only then send it for language editing. It is desirable that the language of the manuscript be corrected by a native English speaker available at our language center and the text be corrected accordingly. Comments are marked in green.

Author Response

The Authors thank the Reviewer for the careful reading and detailed analysis of the manuscript. We were pleased to know that scientifically the manuscript is ok.

Concerning the English. We believe that using “and” and “of” grammatically can be correct.

Many Reviewer’s important comments and suggestions were addressed, corrected and the MS was revised accordingly (marked in yellow).

According to the Reviewer’s recommendation, “of” was removed from the text.

We removed some unnecessary introductory words like “importantly, notably, also”, etc.” from the MS text.

We split the sentences in “Mito ROS” paragraph and also in other paragraphs.

“H2O2, which in turn can easily escape mitochondria” - it was written in the text. Superoxide radical is negatively charged molecule and cannot so easily penetrate through the outer mitochondrial membrane.  

“Also, Wang et al.” Sorry, it is different works and they cannot be merged.

Also, the introductory words (“However,” “For example,” etc.) are widely used in many scientific papers. We think that this could be ok. They create an important link between different sentences, making text flow better readable and easy for readers.

We do not quite understand how we can put introductory part at the sentence end and what adventures it brings. In all scientific papers always the structure is – “Introduction, Results, Discussion”. Therefore, Introduction is always first.

Thank your again.  

Reviewer 2 Report (New Reviewer)

The work presented for review is short and synthetic. In a clear and basic way, it discusses the issue of the production of reactive oxygen species and their importance for the functioning of cells. Apart from a few shortcomings (listed below), it is written correctly and describes the key mechanisms in signalling related to reactive oxygen species. However, a major drawback of this work is the lack of consideration of the latest research and reports on this topic. The work mainly presents knowledge older than 5 or even 10 years. Though the title suggests the interplay between ROS, mitochondria and the entire metabolism is complexed, this work only briefly describes a few aspects of this broad issue. Taking into account the fact that there are many similar reviews on this subject, and the presented work does not analyse the reports from recent years, additionally taking into account such ambiguities as mentioned below the manuscript in the present form is not properly prepared for publication.

1.      Line 70-71 Cu/Zn SOD is mainly cytosolic.

2.      Are mitochondria main ROS producers in all situations/conditions? What about other cellular sources under physiological condition?

3.      Line 98 Is FCCP concentration given by authors general for all types of cells? For both CCCP and FCCP? Or maybe in some situations the concentration might be different for example lower? This needs an example or a reference

4.      The illustration presented in the fig 1 is quite simple, similar to many other schemes in the literature.

5.      Calcium signalling – it is described really superficially, it is not written how the calcium gets into the mitochondria. Nothing about the role of ER in stress response and regulation of mitochondrial functions. The role of PTP is not clear. line 150: is mPTP opened to balance calcium release, so to uptake calcium by mitochondria?

6.      Line 182-183, the sentence about SHC is not clear.

7.      Kinases in ROS signalling – what about JNK kinase? How is PKC regulated by ROS? Is AMPK alone regulating mitochondrial biogenesis or there are other critical factors related to its activity?

8.      What is actually the relevance of mitochondria heterogeneity and what implications it has for certain pathologies? How the subpopulations of mitochondria behave in the face of continuous fusion/fission of mitochondrial network? What is the role of mitophagy?

9.      Line 279 which authors? Reference needed.

Author Response

RC (Reviewer Comment).The work presented for review is short and synthetic. In a clear and basic way, it discusses the issue of the production of reactive oxygen species and their importance for the functioning of cells. Apart from a few shortcomings (listed below), it is written correctly and describes the key mechanisms in signalling related to reactive oxygen species. However, a major drawback of this work is the lack of consideration of the latest research and reports on this topic. The work mainly presents knowledge older than 5 or even 10 years. Though the title suggests the interplay between ROS, mitochondria and the entire metabolism is complexed, this work only briefly describes a few aspects of this broad issue.

Authors Response (AR) The Authors thank the Reviewer for the careful reading and detailed analysis of the manuscript.

  1. In fact, almost each Reviewer’s comment deserves a special, separate large review.

Also, please see the Acknowledgments: - “The authors apologize that they could not cite all-important studies in this field (line 329)”.

  1. Taking into account the fact that there are many similar reviews on this subject, and the presented work does not analyse the reports from recent years,
  2. Please see refs. 14,23,29,38,63,66,89,91,98,104,105,106,115. They are rather new.

  1. Line 70-71 Cu/Zn SOD is mainly cytosolic.

Because mitochondria are the major site of free radical generation, “they are highly enriched with antioxidants including GSH and enzymes, such as superoxide dismutase (SOD) and glutathione peroxidase, on both sides of their membranes to minimize oxidative stress in and around this organelle. The present work reviews the sites and mechanism of ROS generation by mitochondria, mitochondrial localization of Mn-SOD and Cu,Zn-SOD which has been postulated for a long time to be a cytosolic enzyme.

  1. Yes. It has been corrected, but Cu,Zn-SOD may be localized also in mitochondria (e.g. refs. 35,36 and see also M. Inoue et al., Curr Med Chem. 2003, 10(23):2495-505).
  2. RC. Are mitochondria main ROS producers in all situations/conditions? What about other cellular sources under physiological condition?
  3. In the MS, alpha-glycerophosphate-dehydrogenase has been mentioned in the text as another source of ROS (line 94). Also, NADPH oxidases and xanthine oxidase are mentioned in the text as other source of ROS (line 138). ROS production significantly increases under some pathological conditions, for example in ischemi/reperfusion injury.
  4. Line 98 Is FCCP concentration given by authors general for all types of cells? For both CCCP and FCCP? Or maybe in some situations the concentration might be different for example lower? This needs an example or a reference.

  1. This possible cell/tissue specificity is mentioned now in the text (line 98). We mostly used our own experience and we found rather similar optimal FCCP concentrations in various cell types.

  1. The illustration presented in the fig 1 is quite simple, similar to many other schemes in the literature.

  1. Initially, we did not want to include any scheme in our rather short review. It was a special request from the Editorial to add some simple scheme.

  1. Calcium signalling – it is described really superficially, it is not written how the calcium gets into the mitochondria. Nothing about the role of ER in stress response and regulation of mitochondrial functions.

  1. Please see in the Acknowledgments: (line 329) The authors apologize that they could not cite all-important studies in this field.

In the MS text: “Mitochondria also play central role in cellular Ca2+ via interplay between mitochondria, ER and cytosol” (lines 148-164).

New refs were added and discussed here (lines 148-164, refs. 88-91).

  1. The role of PTP is not clear. line 150: is mPTP opened to balance calcium release, so to uptake calcium by mitochondria?

  1. It was written in the MS text: “Under normal conditions, transitions between open and closed states of the mitochondrial permeability transition (MPT) pore operate to balanced/moderate release of mitochondrial Ca2+.” (lines 166-168)

More information and new refs. have been added here.

  1. Line 182-183, the sentence about SHC is not clear.

  1. This has been rephrased.

  1. Kinases in ROS signalling – what about JNK kinase?

  1. More information has been added (lines 220-230). New refs. were added (refs. 105-107). The role of JNK and mitophagy were now mentioned.

  1. How is PKC regulated by ROS?
  2. More information and new refs. were added (lines 220-230, refs. 103-105 ).
  3. Is AMPK alone regulating mitochondrial biogenesis or there are other critical factors related to its activity?
  4. We were mostly focused on the mitochondrial and cell redox metabolism, according to the journal topic “Redox metabolism”. The biogenesis problem is a very large another topic and probably requires another special large review.

  1. What is actually the relevance of mitochondria heterogeneity and what implications it has for certain pathologies? How the subpopulations of mitochondria behave in the face of continuous fusion/fission of mitochondrial network?

It is now mentioned (line 98-100).

  1. For example, different sensitives to ischemia/reperfusion injury has been shown for sub-sarcolemmal (SSM) and intermyofibrillar (IFM) subpopulations, (Lesnefsky et al., ref.22).

See also  E.J. Lesnefsky, C. L. Hoppel. Ischemia-reperfusion injury in the aged heart: role of mitochondria. Arch Biochem Biophys. 2003 Dec 15;420(2):287-97.

Also, Ca2+ and ROS increase  together with MPTP opening may be different in sub-sarcolemmal (SSM) and intermyofibrillar (IFM) subpopulations (which are more resistant to high Ca2+).

  1. What is the role of mitophagy?
  2. It is now mentioned (98-100).

  1. RC. Line 279 which authors? Reference needed.
  2. Ref. has been added (ref. 115).

Round 2

Reviewer 2 Report (New Reviewer)

I thank the authors for considering and taking into account my comments. I appreciate the work authors put into improving the text. However I would like only to comment on the references - according to the authors 13 of 115 references are from last 5 years. It is not a lot, considering the fact that other 22 are older than from year 2000. I appreciate going into the very beggining and original papers in this field however, many discoveries have been made since then e.g. in the field of mitochondrial calcium metabolism, mtPTP role and structure, ROS function. Another downside is that big part of the cited papers are reviews, so actually the authors of the present manuscript are making a review of a reviews, hence it is difficult for me to see what aspect of the presented review is the new approach, what part of the concept is a novelty and was not presented in other reviews on ROS role in metabolism.

Author Response

Rev. #2-Round 2

  1. AR. The Authors thank the Reviewer for careful reading and detailed analysis of the manuscript.
  2. RC. I thank the authors for considering and taking into account my comments. I appreciate the work authors put into improving the text.
  3.  
  4. Thank you very much!
  5.  
  6. RC. However I would like only to comment on the references - according to the authors 13 of 115 references are from last 5 years. It is not a lot, considering the fact that other 22 are older than from year 2000.
  7. AR. No, it was 16: Ref.14 (2019); Ref.23 (2018); Ref. 29 (2021); Ref.38 (2022); Ref.63 (2022); Ref. 65 (2018); Ref. 66 (2019); Ref. 72 (2017); Ref. 89 (2019); Ref. 91 (2018); Ref. 97 (2019); Ref. 98 (2018); Ref. 104 (2020); Ref. 105 (2017); Ref. 106 (2021); Ref. 115 (2021). Concerning other references, we wanted to show a bit about history and the problem development in time, which could be interesting for general readers (even not specialists in mitochondria). Also showing various methods and approaches used in different time.
  1. RC. I appreciate going into the very beginning and original papers in this field however, many discoveries have been made since then e.g. in the field of mitochondrial calcium metabolism, mtPTP role and structure, ROS function.
  2. AR. Again, in the “Acknowledgments”: The authors apologize that they could not cite all-important studies in this field. The ROS function was described in the special paragraph #2. The authors apologize again that not all details were mentioned here. We tried to make it compact and easy for general readers. Also, “… mitochondrial calcium metabolism, mtPTP role and structure were not major topics of this review and certainly can deserve special separate large reviews (which are already exist and some, we think most important, are cited here).  
  1. RC. Another downside is that big part of the cited papers are reviews, so actually the authors of the present manuscript are making a review of a reviews, hence it is difficult for me to see what aspect of the presented review is the new approach, what part of the concept is a novelty and was not presented in other reviews on ROS role in metabolism.
  2. AR. Usually, reviews summarize already published discoveries and not something absolutely new (approaches, methods, researches, etc.) as well as novelty and concepts not presented elsewhere, this is prerogative of original articles (not reviews). We thought, and the idea was, to write something reliable and interesting for general readers, even not specialists in the field. If somebody will want to see some aspects more deeply, he can easily find them in reviews, contained usually 100-200 references.

Thank you again.

This manuscript is a resubmission of an earlier submission. The following is a list of the peer review reports and author responses from that submission.

Round 1

Reviewer 1 Report

In this manuscript Kuznetsov and colleagues provide a general review on mitochondrial ROS, Ca2+ signalling, their relationship with kinase signalling, mitochondrial environment and comment about imaging of the mitochondrial redox state. While the review is interesting, there are a few comments which have to be addressed before its publication in Antioxidants.

Major comments:

-       Lines 73-74: “by mitochondrial manganese superoxide dismutase (MnSOD), which in turn can be scavenged by the enzyme catalase to water.”. First, Mitochondrial not only contain MnSOD, but also Cu,ZnSOD in the inner mitochondrial membrane (IMM). Second, catalase is an enzyme found exclusively in peroxisomes; H2O2 detoxifying enzymes are peroxiredoxin-3 (Prx3), Prx5 or Glutaredoxin 2 (Grx2). All these must be corrected.

-       Line 83: “Mitochondrial ROS, particularly under pathological conditions, can damage several cellular elements such as DNA (and mitochondrial DNA), various proteins and enzymes (especially by oxidation of sensitive and critical their –SH groups)”. Oxidation of thiol groups can be used for signalling or be toxic. This depends usually of the reversibility of reaction; thus, when sulfenic acid is form, which is a reversible thiol modification, it has a signalling purpose. However, sulfinic and sulfonic forms are irreversible and usually toxic. Also, a very toxic modification catalyzed by ROS is the release of iron from Fe-S clusters in enzymes. All these should be included in the text.

-       In the section: “ROS-induced ROS release and Mitochondrial ROS Flashes”, caution must be applied when talking about mitochondrial flashes due to the fact that cpYFP not only presumably detects superoxide anion, but also pH (PMID: 27463140; 25341790). Thus, this section must contain a paragraph discussing the the nature of the mitoflashes, mostly regarding the specificity of the main tool used to detect them (cpYFP) and whether the physiological events reviewed may be caused by changes in mitochondrial pH, ROS orboth.

-       In the section: “The interplay between ROS and Ca2+ signaling” the authors relate mitochondrial Ca2+ to ROS production and exclusively discuss rather old literature. More recent advances in the field should be added and discussed: PMID: 32728214, 31467276, 21068725.

-       How is that Mitotracker Green can be colocalized with flavoproteins when they have the same excitation/emission spectra? This should be commented or corrected.

-       Fig.1 shows yellow (=red) mitochondrial staining, but also red nucleoli staining. I wonder whether the cell were stained with MitoSOX instead of Rhod-2? Please, re-check.

-       Line 91: “An important controller of mitochondrial superoxide generation rates is the mitochondrial inner membrane potential (Δψm), and uncouplers like 2,4-dinitrophenol or FCCP and the uncoupling proteins (UCPs) may decrease mitochondrial ROS production”. This only occurs under very specific conditions enabling complex I reverse electron transport (RET). There are many conditions in which FCCP or DNP enhance superoxide production by the electron transport chain. This should be corrected or mentioned.

Minor comments:

-       Line 49: O2 should be written with subscript.

-       Rephrase sentence in lines 108-110. It is difficult to follow.

-       Lines 127-128: Correct last sentence in paragraph.

-       Line 180: Correct “recognized tzo be

-       The phrase: “These both fluorescence emissions can be used for the sensitive assessment of the region-specific mitochondrial redox state.” need references.

-       Line 222: correct “depend of

-       Sentence: “Importantly, flavoproteins and NADH fluorescence were fully co-localized with MitoTracker Green, well established fluorescent marker for mitochondria.” require references.

Reviewer 2 Report

This review attempts to describe how cellular metabolic changes and associated signaling pathways regulate the bidirectional link between energy production and Ca kinetics and ROS production, which are the primary functions of mitochondria.

The review should indicate what is currently known, what has been elucidated and agreed upon, and what needs to be considered in the future. Despite the vast scope of the subject matter covered and the vast amount of research results, the review does little to provide an overview of the current state of the art, and simply uses the term "complex relationships," which is likely to cause great frustration to the reader.

In addition, the topics covered are fragmented and not sufficiently described to promote a comprehensive understanding.

Although a methodology for observing redox states is described, I believe that the description of a methodology developed by the authors that is suddenly unrelated to the main objective, the interconnection of energy, Ca kinetics, and reactive oxygen species, is inappropriate and only serves to blur the main focus of this review.